# Effects of Fumarate and Nitroglycerin on In Vitro Rumen Fermentation, Methane and Hydrogen Production, and on Microbiota

**DOI:** 10.3390/biology12071011

**Published:** 2023-07-15

**Authors:** Jichao Li, Shengwei Zhao, Zhenxiang Meng, Yunlong Gao, Jing Miao, Shengyong Mao, Wei Jin

**Affiliations:** 1Ruminant Nutrition and Feed Engineering Technology Research Center, College of Animal Science and Technology, Nanjing Agricultural University, Nanjing 210095, China; ljcsci@126.com (J.L.); 2020105048@stu.njau.edu.cn (S.Z.); c86818629@126.com (Z.M.); zz38166498@126.com (Y.G.); miaojing@njau.edu.cn (J.M.); maoshengyong@njau.edu.cn (S.M.); 2Laboratory of Gastrointestinal Microbiology, College of Animal Science and Technology, Nanjing Agricultural University, Nanjing 210095, China

**Keywords:** nitroglycerin, fumarate, rumen microbiota, rumen fermentation, methanogenesis

## Abstract

**Simple Summary:**

An important strategy to mitigate global warming is to reduce methane produced by ruminants. However, the inhibition of rumen methanogenesis generally results in hydrogen accumulation, which would affect the normal fermentation in the rumen. In this study, we used a combination of two chemicals (fumarate and nitroglycerin) to mitigate the rumen methane production. Nitroglycerin inhibits the activities of methanogens. Fumarate eliminates hydrogen accumulation. In vitro rumen fermentation was used to investigate the effects of this combination on rumen fermentation, methane and hydrogen production, and microbiota. The results showed that the addition of fumarate decreased the hydrogen accumulation and increased the concentration of propionate and microbial crude protein when methanogen activities were inhibited by nitroglycerin. The bacterial and archaeal communities were altered by the addition of the two chemicals, with several taxa changed in the relative abundance. Conclusively, the combination of fumarate and nitroglycerin inhibited methane production, reduced hydrogen accumulation, improved rumen fermentation and altered rumen microbiota. This study provides an alternative way of using these chemicals in order to mitigate methane emission in ruminants.

**Abstract:**

This study aimed to investigate the effects of fumarate and nitroglycerin on rumen fermentation, methane and hydrogen production, and microbiota. In vitro rumen fermentation was used in this study with four treatment groups: control (CON), fumarate (FA), nitroglycerin (NG) and fumarate plus nitroglycerin (FN). Real-time PCR and 16S rRNA gene sequencing were used to analyze microbiota. The results showed that nitroglycerin completely inhibited methane production and that this resulted in hydrogen accumulation. Fumarate decreased the hydrogen accumulation and improved the rumen fermentation parameters. Fumarate increased the concentration of propionate and microbial crude protein, and decreased the ratio of acetate to propionate in FN. Fumarate, nitroglycerin and their combination did not affect the abundance of bacteria, protozoa and anaerobic fungi, but altered archaea. The PCoA showed that the bacterial (Anosim, R = 0.747, *p* = 0.001) and archaeal communities (Anosim, R = 0.410, *p* = 0.005) were different among the four treatments. Compared with CON, fumarate restored Bacteroidetes, Firmicutes, Spirochaetae, Actinobacteria, Unclassified *Ruminococcaceae*, *Streptococcus*, *Treponema* and *Bifidobacterium* in relative abundance in FN, but did not affect *Succinivibrio*, *Ruminobacter* and archaeal taxa. The results indicated that fumarate alleviated the depressed rumen fermentation caused by the inhibition of methanogenesis by nitroglycerin. This may potentially provide an alternative way to use these chemicals to mitigate methane emission in ruminants.

## 1. Introduction

Methane (CH_4_) is a major contributor to global climate change [1]. About 40% of greenhouse gas emissions from livestock production can be attributed to ruminant CH_4_ production, which accounts for approximately 6% of global anthropogenic greenhouse gas emissions [2]. Ruminal CH_4_ production does not only concern greenhouse gas emissions, but also relates to energy loss for ruminants (up to 12% of the total energy intake) [3]. The rumen is rich in bacteria, protozoa, fungi and methanogens, which can ferment coarse feedstuffs to produce volatile fatty acids, carbon dioxide and CH_4_. Hydrogen (H_2_) is an important intermediate in most of those biochemical processes [4]. CH_4_ is generally produced through the utilization of CO_2_ and H_2_ by methanogenic archaea in the rumen [5]. The accumulation of H_2_ could affect the normal rumen fermentation, therefore, CH_4_ generation plays an important role in H_2_ elimination in the rumen.

Strategies like the use of feed additives, nutrition management and animal genetic improvement have been proposed for use in reducing CH_4_ emissions from ruminants [6,7,8]. Chemicals such as sulfate, nitrate and fumarate were studied for their potential to reduce rumen CH_4_ emissions [9,10,11]. The CH_4_ inhibitor 3-nitrooxypropanol [12,13] and the macroalga *Asparagopsis taxiformis* [14,15] were recently developed as promising rumen CH_4_-mitigating agents. However, the inhibition of methanogenic activity usually results in abnormal rumen fermentation caused by H_2_ accumulation. Therefore, it is necessary to devise an alternative means of eliminating H_2_ when inhibiting the activity of methanogens. Fumarate is a metabolic intermediate and can be reduced to succinate by H_2_ in the rumen. Succinate is then decarboxylated into propionate, which is a major energy source for ruminants [16]. Fumarate is a promising H_2_-comsuming chemical in the rumen. Nitroglycerin, targeting methanogens, has the same functional group as the chemical 3-nitrooxypropanol, which was reported as being an effective means of reducing ruminal CH_4_ emissions in in vitro and in vivo studies [17,18]. Its metabolic end products are propionate and ammonia in the rumen.

This study hypothesized that fumarate would alleviate the abnormal rumen fermentation when methanogenesis was inhibited by nitroglycerin. The objective of this study was to investigate the effects of fumarate and nitroglycerin on rumen fermentation, CH_4_ and H_2_ production and microbiota in an in vitro rumen trial. The results of this study could help to develop an alternative means of mitigating rumen CH_4_ emissions.

## 2. Materials and Methods

### 2.1. Animals

Three rumen-fistulated Chinese Hu sheep were fed on a maintenance diet for a period of 30 days. On day 31, 500 mL of rumen fluid was collected 2 h before the sheep were fed. The diet of the 3 sheep was prepared in accordance with the maintenance requirements (NY/Y 816-2004; Ministry of Agriculture of China, 2004), including 70% Chinese wild rye, 20% corn, 7% soybean meal, 1.5% CaHPO_4_, 0.5% stone powder, 0.5% NaCl and 0.5% additives (Vitamin and mineral mix contained the following ingredients per kilogram of diet: vitamin A, 22.5 KIU/kg; vitamin D3, 5.0 KIU/kg; vitamin E, 37.5 IU/kg; vitamin K3, 5.0 mg/kg; Mn, 63.5 mg/kg; Zn, 111.9 mg/kg; Cu, 25.6 mg/kg; and Fe, 159.3 mg/kg), and comprised 94.01% dry matter, 10.01% crude protein, 2.39% ether extract, 51.48% neural detergent fiber, 30.94% acid detergent fiber and 7.73% crude ash on a dry-matter basis. Sheep were fed a total mixed ration twice daily (08:00 and 17:00) and had free access to fresh water.

### 2.2. Experimental Design

In vitro rumen fermentation was carried out with a completely randomized design (CRD) for 4 treatments: control (CON), fumarate at 12 mmol/L (FA), nitroglycerin at 99 μmol/L (NG) and fumarate at 12 mmol/L plus nitroglycerin at 99 μmol/L (FN). The dosages of FN and NG used in this study were determined according to previous studies [19,20] and the results of the pre-experiments. Three replicates were prepared for each treatment. Additionally, three independent incubation runs were performed at different times [21]. Each run consisted of 4 treatments with 3 replicates and 4 blanks containing only the inoculum. The experimental procedure was conducted according to the study of Martínez-Fernández et al. [22]. The rumen fluid was collected from 3 Hu sheep before the morning feeding was performed, and this fluid was then pooled and filtered through 4 layers of cheesecloth. The filtered rumen fluid and buffer were mixed thoroughly (1:3 [*vol*/*vol*]) in a water bath at 39 °C under anaerobic conditions. Additionally, each 100 mL of the mixture was dispensed into a 180 mL serum bottle containing 1.0 g substrate and chemicals (fumarate or nitroglycerin). All serum bottles were sealed and incubated at 39 °C for 24 h at 80 rpm. After 24 h of incubation, all the fermentation flasks were taken out and put into ice water to terminate the fermentation. Samples were collected and stored for subsequent analysis. The buffer was composed of 8.75 g NaHCO_3_, 1.00 g NH_4_HCO_3_, 1.43 g Na_2_HPO_4_, 1.55 g KH_2_PO_4_, 0.15 g MgSO_4_·7H_2_O, 0.52 g Na_2_S, 0.017 g CaCl_2_·2H_2_O, 0.015 g MnCl_2_·4H_2_O, 0.002 g CoCl_2_·6H_2_O, 0.012 g FeCl_3_·6H_2_O and 1.25 mg resazurin per liter [23]. The composition of the substrate was the same as the diet provided to sheep. The substrate was dried at 65 °C for 48 h and passed through a 1 mm screen with a Wiley mill (Arthur H. Thomas, Philadelphia, PA, USA). The fumarate (disodium fumarate) was purchased from Shanghai Macklin Biochemical Co., Ltd. (Shanghai, China). Nitroglycerin was purchased from Beijing Yimin Pharmaceutical Co., Ltd. (Beijing, China).

### 2.3. Sample Collection and Chemical Analysis

At the end of the fermentation, the pH value was measured using a pH meter (Ecoscan pH 5, Thermo Fisher Scientific Inc., Singapore). Then, the bottles were immediately put into ice water to stop fermentation. The supernatant of the fermentation fluid was collected and stored at −20 °C for the determination of volatile fatty acids (VFAs), lactate, microbial crude protein (MCP) and ammonia nitrogen (NH_3_-N). The mixture of the substrate and fermentation fluid was collected and stored at −80 °C for the analysis of the microbiota.

### 2.4. Sample Collection and Chemical Analysis

Gas production was assessed using a pressure transducer [24]. Methane (CH_4_) and hydrogen (H_2_) production were measured following the gas measurement procedure using a GC-TCD instrument (Agilent 7890B, Agilent Technologies Inc., Santa Clara, CA, USA). Gases were separated on packed GC columns (Porapak Q packing & MolSieve 5A packing, Agilent Technologies Inc., CA, USA) at a column temperature of 80 °C, a 200 °C injection temperature and a 200 °C TCD detector temperature. N_2_ was the carrier gas. The VFAs were determined according to Jin et al. [17]. Each 1.0 mL sample was mixed with 0.2 mL deproteinization–acidification solution [metaphosphoric acid (25% *w*/*v*) and crotonic acid (0.65% *w*/*v*)] before undergoing analysis via gas chromatography (Agilent 7890B instrument, Agilent Technologies Inc., CA, USA). The sample was separated using a fused silica capillary column (Supelco, Bellefonte, PA, USA) with a programmed heating process (110 °C for 3 min, 110–150 (40 °C/min)). The injection temperature was 200 °C. The flame ionization detector temperature was 220 °C. The carrier gas was nitrogen. Lactate was measured using an assay kit in accordance with the instructions of the manufacturer (Jiancheng Bioengineering Research Institute, Nanjing, China). Microbial crude protein was determined with a commercial reagent kit (BCA Protein Assay Kit, Tiandz Inc., Beijing, China) in accordance with manufacturer instructions. The concentration of NH_3_-N was analyzed using an indophenol method with an acidified procedure [25].

### 2.5. DNA Extraction and Real-Time PCR

Genomic DNA was extracted from a 1.0 mL sample using a bead-beating and phenol-chloroform–isopentanol extraction method [26]. Each DNA sample was divided into two parts to perform sequencing and real-time PCR.

Archaea, bacteria, anaerobic fungi and protozoa were quantified using an Applied Biosystems 7300 Real time PCR system (Applied Biosystems, Thermo Fisher Scientific Inc., Foster City, CA, USA). The primers for the 4 microbial populations are listed in Table A1. An SYBR^®^ Premix Ex Tag TM (TaKaRa, Dalian, China) was used to prepare the reaction mixture. The copy number of DNA in each sample was measured in triplicate, and the average value was calculated. The external standards were prepared with plasmid DNA of clones of each microbial population. The results are expressed as the number of copies of marker genes per milliliter of fermentation liquid.

### 2.6. 16S rRNA Gene Sequencing and Data Analysis

The 16S rRNA genes of bacteria were amplified with a primer pair 341F (5′-CCTAYGGGRBGCASCAG-3′) and 806R (5′-GGACTACNNGGGTATCTAAT-3′). The 16S rRNA genes of archaea were amplified with a primer pair Met86F (5′-GCTCAGTAACACGTGG-3′) and Met471R (5′-GWRTTACCGCGGCKGCTG-3′). The amplicons were subjected to double-ended sequencing (paired sequencing) using an Illumina MiSeq PE250 platform produced by BIOZERON Biotechnology Co., Ltd. (Shanghai, China). The raw data were stored in the Sequence Read Archive (SRA) database of the National Biotechnology Information Center (NCBI) (accession number: PRJNA913631, bacteria; PRJNA913641, archaea).

Fastp (version 0.20.0) and FLASH (version 1.2.7) were used to filter and merge 16S rRNA sequences, and the chimeras were filtered to obtain effective reads [27]. UPARSE (version 7.1) was used to pick up the operational taxonomic unit (OTU) with a 97% similarity truncation value [28]. Taxonomic assignment was performed for bacteria using RDP classifier (version 2.11) based on the SILVA database (version 138), and via the RIM-DB database for methanogens [18]. QIIME 2 was used for alpha diversity analysis. The principal coordinate analysis (PCoA) was conducted based on Bray–Curtis distance [29]. The significance of the differences among groups was assessed with ANOSIM using the vegan package in R.

### 2.7. Statistical Analysis

The analyses of the in vitro fermentation parameters and the real-time PCR data were performed using the MIXED procedure of SAS 9.4 version (SAS Institute, Inc., Cary, NC, USA), and the data were tested to determine their normality using the Shapiro–Wilk test of SAS. The model used for data analysis was *Yijk* = *μ* + *P_i_* + *S_j_* + *PS_ij_* + *e_ij_*, where *Y_ijk_* is the observed value, *μ* is the overall mean, *P_i_* is the fixed effect of treatment with nitroglycerin, *S_j_* is the fixed effect of treatment with fumarate, *PS_ij_* is the interaction effect of nitroglycerin * fumarate and *e_ij_* is the random error. The variables that had non-normal distributions were analyzed using the Kruskal–Wallis test procedure. The Tukey test was used to identify differences (*p* < 0.05) between means.

## 3. Results

### 3.1. Total Gas, Hydrogen and Methane Production

The total gas production in FA was the highest among the four groups (*p* < 0.05, Table 1), and it was higher in FN than CON and NG (*p* < 0.05). Hydrogen was accumulated in NG and FN. Additionally, NG had the highest hydrogen production (*p* < 0.05). Methane was only accumulated in CON and FA. There was no methane detected in NG and FN. Methane production was higher in FA than CON (*p* < 0.05).

### 3.2. Fermentation Characteristics

The in vitro fermentation characteristics are presented in Table 2. Total VFA was only higher in FA than NG (*p* < 0.05). Acetate was higher in CON and FA than NG and FN (*p* < 0.05). Propionate was higher in FA and FN than CON and NG (*p* < 0.05). The ratio of acetate/propionate in CON was the highest (*p* < 0.05), and it was higher in FA and NG than FN (*p* < 0.05). Isobutyrate in FA was the highest (*p* < 0.05), and it was higher in CON than NG (*p* < 0.05). Valerate was higher in NG and FN than CON and FA (*p* < 0.05). Isovalerate in FA was the highest (*p* < 0.05), and it was higher in CON than NG and FN (*p* < 0.05). Ammonia nitrogen in FA was the highest (*p* < 0.05). Microbial crude protein was higher in CON and FA than NG and FN (*p* < 0.05), and it was higher in FN than NG (*p* < 0.05). There were no significant differences in pH, butyrate and lactate among the four groups (*p* > 0.05).

### 3.3. The Quantification of Protozoa, Bacteria, Anaerobic Fungi and Archaea

There was no significant difference in the abundance of bacteria, protozoa and anaerobic fungi (*p* > 0.530) among the four groups (Table 3). The abundance of archaea was higher in FA than CON and NG (*p* < 0.05), and it was higher in FN than NG (*p* < 0.05). 

### 3.4. Bacterial Community

A total of 390,011 bacterial sequences remained after filtering for quality. The average length was 418 bp. A total of 2722 OTUs were identified. There was no difference in the alpha diversity indexes (*p* > 0.05, Table 4), except for the Shannon index (*p* = 0.043). There was a clear separation of clusters on the 3D-PCoA plot of bacterial populations among the four groups (Anosim, R = 0.747, *p* = 0.001, Figure 1A). PC1, PC2 and PC3 accounted for 53.08%, 16.04% and 10.51% of the total variance, respectively.

At the phylum level, 20 phyla were identified across all samples. The eight predominant phyla (the average relative abundances of phyla >1% in at least one group) were Bacteroidetes, Firmicutes, Proteobacteria, Spirochaetes, Actinobacteria, Candidate_division_SR1, Candidate_division_TM7 and Cyanobacteria (Table 5). The relative abundance of Bacteroidetes was higher in FN than NG (*p* < 0.05), but it did not show significant differences between other groups (*p* > 0.05). Firmicutes and Actinobacteria were the highest in NG (*p* < 0.05). Proteobacteria in CON and FA were higher than NG and FN (*p* < 0.05). Spirochaetae were higher in FA and FN than NG (*p* < 0.05). Candidate_division_TM7 was the highest in CON (*p* < 0.05). The relative abundance of Cyanobacteria showed no significant difference among groups (*p* > 0.05).

A total of 307 bacterial genera were identified from all samples. The seven predominant genera (the average relative abundances of genera >2% in at least one group) were *Streptococcus*, *Succinivibrio*, *Ruminobacter*, *Treponema*, *Bifidobacterium*, Unclassified *BS11_gut_group* and Unclassified *Ruminococcaceae* (Table 6). The relative abundance of *Streptococcus* and *Bifidobacterium* was the highest in NG (*p* < 0.05). *Succinivibrio* was higher in CON and FA than FN (*p* < 0.05). *Ruminobacter* was higher in CON than NG and FN (*p* < 0.05). *Treponema* was higher in FN than NG (*p* < 0.05). Unclassified *BS11_gut_group* was higher in CON and FA than the other two groups (*p* < 0.05). Unclassified *Ruminococcaceae* was lower in NG than CON (*p* < 0.05). 

### 3.5. Archaeal Community

A total of 568, 155 archaeal sequences were obtained after quality filtering. The average length was 355 bp. No difference in the alpha diversity index was found (*p* > 0.05, Table 4). These sequences were clustered into 198 OTUs. There was a clear separation of clusters on the 3D-PCoA plot of archaeal populations among the four groups (Anosim, R = 0.410, *p* = 0.005, Figure 1B). PC1, PC2 and PC3 accounted for 73.62%, 8.94% and 5.96% of the total variance, respectively. 

A total of two archaeal orders were identified from all samples. The two predominant orders (the average relative abundance of orders > 1% in at least one group) were Methanobacteriales and Methanomassiliicoccales (Table 7). The relative abundance of Methanobacteriales was the lowest in FN (*p* < 0.05), and it was lower in FA than CON or NG (*p* < 0.05). Additionally, there was no significant difference between CON and NG (*p* > 0.05). The relative abundance of Methanomassiliicoccales was the highest in FN (*p* < 0.05), and it was higher in FA than CON and NG (*p* < 0.05).

A total of 47 archaeal species were identified. Six predominant species (the average relative abundance of species > 1% in at least one group) are shown in Table 8. The relative abundance of *Methanobrevibacter gottschalkii clade* was lower in FN than CON (*p* < 0.05). *Group12* sp. *ISO4-H5* was the highest in FN (*p* < 0.05). *Group9* sp. *ISO4-G1* was higher in FN than the other groups (*p* < 0.05), and it was higher in FA than CON and NG (*p* < 0.05).

## 4. Discussion

The inhibition of methanogenic activities usually results in H_2_ accumulation and causes depressed rumen fermentation [30]. This can affect the animal production performance. The results of this study showed that the addition of fumarate alleviated H_2_ accumulation and improved the depressed rumen fermentation parameters when methanogenesis was inhibited by nitroglycerin. Nitroglycerin was demonstrated to be effective at reducing rumen methane production in several in vitro and in vivo studies [17,18,20]. It was able to completely inhibit methane production and caused an accumulation of hydrogen in in vitro rumen fermentation [17,20]. Moreover, the final metabolites of nitroglycerin were propionate and ammonia in the rumen, which have no negative effect on rumen fermentation. Therefore, nitroglycerin was used to successfully establish a model of methane depression and hydrogen accumulation in this study. Nitroglycerin caused about 4.8% hydrogen of accumulation (% total gas production). The current experiment also observed that nitroglycerin inhibited methane production, but did not affect the abundance of archaea compared with the control. This result is consistent with that of a previous study [20]. In another study, an opposite result was observed, whereby the abundance of archaea declined when the methane production was depressed by nitroglycerin [17], which was consistent with the research on the other methanogenic inhibitors [31,32]. However, the mechanisms of the different results for archaeal abundance in different studies are unclear and further work is needed to elucidate this point.

Fumarate is an intermediate in the rumen metabolism and is finally reduced to propionate [33]. The reduction of fumarate has a lower H_2_-consuming threshold (0.02 ppm) and produces more Gibbs free energy than the methanogenesis of H_2_ and CO_2_. The fumarate reduction should be more effective than methanogenesis in the rumen [34,35,36]. Therefore, fumarate was used as a rumen CH_4_-mitigation agent in many previous studies [37,38,39]. However, the effects of fumarate on rumen CH_4_ mitigation were found to be inconsistent. Bayaru et al. [37] observed that CH_4_ production in steers was reduced by 23% when fumarate was added to the complete diet at 20 g/kg dry matter. In contrast, no effect was observed in steers fed barley silage and concentrate with fumarate (12 g/kg dry matter) [38] and in lambs fed dried alfalfa with fumarate (100 g/kg dry matter) [39]. Fumarate increased CH_4_ production in sheep fed a mixed diet [19], which is consistent with the current study. Fumarate increased the abundance of archaea, methane production and acetate with the absence of nitroglycerin. Fumarate was expected to consume H_2_ and reduce methane production; the increase in methane production was not expected. Fumarate can be metabolized into acetate via the malate–pyruvate pathway in the rumen [19]. In this process, there is net [H] produced (C_4_H_4_O_4_ + 2H_2_O→C_2_H_4_O_2_ + 2CO_2_ + 4H), which could account for the increase in the abundance of archaea and methane production. The increased concentration of acetate supported this speculation. Moreover, Gibbs free energy calculation shows that the production of acetate from fumarate under rumen conditions is thermodynamically feasible even at very low fumarate concentrations [35]. Fumarate was metabolized into acetate instead of propionate, which could have occurred as the microbial populations that reduce fumarate to succinate/propionate had not yet been completely established. It may also give an explanation for the inconsistent results obtained in different studies on fumarate. 

Fumarate and nitroglycerin altered the relative abundance of Bacteroides and Firmicutes as well as *Streptococcus* and several unclassified genera to the two phyla. The relative abundance of *Streptococcus* was increased by nitroglycerin, but restored to the level of that in CON by the addition of fumarate. The underlying mechanism of the changes in *Streptococcus* is unclear. However, *Streptococcus* had been reported to produce bacteriocin, which could inhibit methane production [40]. *Succinivibrio*, belonging to Proteobaceria, produces succinate [41]. The relative abundance of *Succinivibrio* was decreased by the combination of fumarate and nitroglycerin. Mao et al. [42] reported that the relative abundance of *Succinivibrio dextrinisolvens* was increased in the rumen of goats fed disodium fumarate. It seems that the combination of the two chemicals had an opposite impact on *Succinivibrio*. *Treponema*, belonging to Spirochaetae, produces succinate, formate and acetate [43]. The relative abundance of *Treponema* was restored by the addition of fumarate. Jin et al. [44] observed an increase in *Treponema* due to disodium fumarate in an in vitro rumen fermentation. Therefore, *Treponema* might play a role in the restoration of the propionate concentration in FN. *Ruminobacter*, belonging to Proteobacteria, is associated with ruminal fiber degradation [45]. Nitroglycerin decreased the proportion of *Ruminobacter*, suggesting the inhibition of fiber degradation. It might be partly related to the decrease in acetate concentration caused by nitroglycerin. *Bifidobacteria*, belonging to the phylum of Actinobacteria, were known fermenters of starch and simple sugars [46]. The addition of nitroglycerin increased the relative abundance of *Bifidobacteria*, but fumarate restored it. The mechanism underlying the changes in *Bifidobacteria* is unclear.

The most dominant methanogens belonged to Methanobacteriales and Methanomassiliicoccales, which was consistent with the results of previous studies [17,18]. Fumarate decreased the relative abundance of Methanobacteriales and increased that of Methanomassiliicoccales. Members of the Methanobacteriales primarily use H_2_ and CO_2_ to produce CH_4_, which generates lower amounts of Gibbs free energy and has a higher H_2_-utilizing threshold than fumarate reduction [35]. The fumarate reduction might have decreased the H_2_ concentration, which depressed the growth of Methanobacteriales. Members of the Methanomassiliicoccales are H_2_-dependent methyltrophic methanogens which produce more Gibbs free energy and have a lower H_2_-utilizing threshold than Methanobacteriales. Moreover, the repair system of Methanomassiliicoccales seems to be more resilient than that of Methanobacteriales in the presence of nitroglycerin [47]. This might explain the changes in the relative abundance in the two methanogenic orders.

## 5. Conclusions

Fumarate decreased hydrogen accumulation and increased the concentration of propionate and MCP in the presence of nitroglycerin. Treatments did not affect the abundance of bacteria, protozoa and anaerobic fungi, but altered the abundance of archaea. Fumarate restored the relative abundance of several bacterial taxa to the levels in CON in the presence of nitroglycerin, but did not affect *Succinivibrio*, *Ruminobacter* and archaeal taxa. Collectively, fumarate alleviated the depressed rumen fermentation caused by the inhibition of methanogenesis by nitroglycerin. This might provide an alternative way to use those chemicals to mitigate methane emissions in ruminants. However, further studies are needed in order to evaluate the effects of nitroglycerin combined with fumarate on animal health, production performance, rumen fermentation and the microbial community in vivo.

## Figures and Tables

**Figure 1 biology-12-01011-f001:**
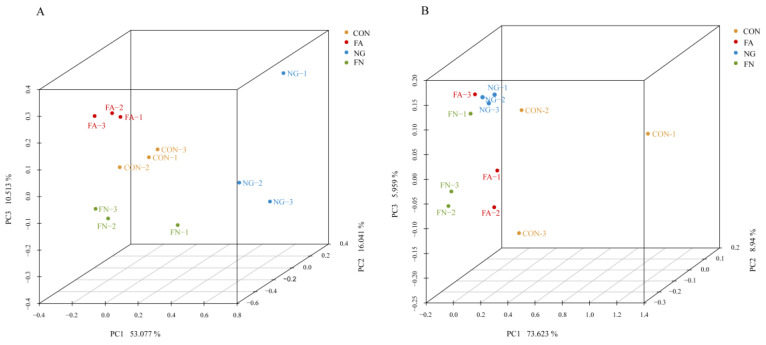
(**A**), 3D-PCoA analysis of bacterial populations based on Bray–Curtis distance, Anosim (R = 0.747, *p* = 0.001); (**B**), 3D-PCoA analysis of archaeal populations based on Bray–Curtis distance, Anosim (R = 0.410, *p* = 0.005).

**Table 1 biology-12-01011-t001:** Gas production from 24 h in vitro fermentation.

Items	Groups ^1^				SEM		*p*-Value	
	CON	FA	NG	FN		FA	NG	NG * FA
Total gas (mL)	203.67 ^c^	245.67 ^a^	201.67 ^c^	226.67 ^b^	5.46	<0.001	<0.001	<0.001
Hydrogen (mL)	0.06 ^c^	0.11 ^c^	9.68 ^a^	6.38 ^b^	1.25	<0.001	<0.001	<0.001
Methane (mL)	21.30 ^b^	24.07 ^a^	0.00 ^c^	0.00 ^c^	3.44	0.020	<0.001	0.020

^a–c^ Means within a row with different superscripts differ (*p* < 0.05). ^1^ CON, control group; FA, 12 mmol/L fumarate was added; NG, 99 μmol/L nitroglycerin was added; FN, fumarate and nitroglycerin were added in combination (FA + NG, 12 mmol/L + 99 μmol/L). * Represents an interactive effect.

**Table 2 biology-12-01011-t002:** Fermentation parameters from 24 h in vitro fermentation.

Items	Groups ^1^				SEM		*p*-Value	
	CON	FA	NG	FN		FA	NG	NG * FA
pH	6.63	6.34	6.21	6.59	0.03	0.053	0.848	0.187
Total VFA (mmol/L)	60.65 ^ab^	73.63 ^a^	45.32 ^b^	60.01 ^ab^	3.44	0.007	0.005	0.829
Acetate (mmol/L)	32.88 ^a^	36.87 ^a^	20.82 ^b^	25.06 ^b^	2.08	0.073	<0.001	0.950
Propionate (mmol/L)	19.07 ^b^	26.61 ^a^	15.34 ^b^	23.93 ^a^	1.40	<0.001	0.026	0.665
A:P	1.72 ^a^	1.39 ^b^	1.36 ^b^	1.05 ^c^	0.08	0.001	<0.001	0.846
Butyrate (mmol/L)	6.79	7.66	5.80	8.13	0.65	0.280	0.855	0.612
Isobutyrate (mmol/L)	0.30 ^b^	0.48 ^a^	0.06 ^c^	0.19 ^bc^	0.03	0.001	<0.001	0.354
Valerate (mmol/L)	1.24 ^b^	1.33 ^b^	3.30 ^a^	2.63 ^a^	0.17	0.192	<0.001	0.099
Isovalerate (mmol/L)	0.37 ^b^	0.69 ^a^	0.01 ^c^	0.07 ^c^	0.05	<0.001	<0.001	0.001
N-NH_3_ (mol/L)	10.68 ^b^	12.62 ^a^	9.46 ^b^	9.52 ^b^	0.20	0.012	<0.001	0.017
MCP (mg/dL)	3.33 ^a^	3.35 ^a^	2.99 ^c^	3.16 ^b^	0.03	<0.001	<0.001	<0.001
Lactate (mmol/L)	0.81	0.63	0.72	0.71	0.02	0.258	0.988	0.276

^a–c^ Means within a row with different superscripts differ (*p* < 0.05). ^1^ CON, control group; FA, 12 mmol/L fumarate was added; NG, 99 μmol/L nitroglycerin was added; FN, fumarate and nitroglycerin were added in combination (FA + NG, 12 mmol/L + 99 μmol/L). Total VFA = total volatile fatty acid; A: P = acetate/propionate; NH_3_-N = ammonia nitrogen; MCP = microbial crude protein. * Represents an interactive effect.

**Table 3 biology-12-01011-t003:** The abundance of protozoa, bacteria, anaerobic fungi and archaea after 24 h in vitro fermentation.

Items	Groups ^1^				SEM		*p*-Value	
	CON	FA	NG	FN		FA	NG	NG * FA
Bacteria (log_10_/L)	9.31	9.32	9.48	9.42	0.03	0.647	0.053	0.538
Protozoa (log_10_/L)	6.55	6.14	6.01	6.85	0.16	0.488	0.778	0.063
Anaerobic fungi (log_10_/L)	6.41	6.82	6.73	6.74	0.10	0.336	0.567	0.351
Archaea (log_10_/L)	7.93 ^bc^	8.35 ^a^	7.64 ^c^	8.06 ^ab^	0.09	0.001	0.011	0.971

^a–c^ Means within a row with different superscripts differ (*p* < 0.05). ^1^ CON, control group; FA, 12 mmol/L fumarate was added; NG, 99 μmol/L nitroglycerin was added; FN, fumarate and nitroglycerin were added in combination (FA + NG, 12 mmol/L + 99 μmol/L). * Represents an interactive effect.

**Table 4 biology-12-01011-t004:** Alpha diversity of bacterial and archaea populations.

Items	Groups ^1^				SEM	*p*-Value
	CON	FA	NG	FN		
Bacterial						
Reads	31,504.7	33,801.3	32,442.3	32,255.3	477.6	0.439
Coverage	0.99	0.99	0.99	0.99	0.00	0.532
Chao 1	2108.0	2208.0	1723.0	2052.0	85.39	0.144
Shannon	6.27 ^a^	6.17 ^a^	5.36 ^b^	6.16 ^a^	0.14	0.043
Simpson	0.01	0.01	0.05	0.01	0.01	0.063
Archaea						
Reads	50,258.0	54,952.0	34,009.0	50,166.0	3133.5	0.060
Coverage	0.98	0.98	0.97	0.98	0.00	0.006
Chao 1	2744.0	2870.0	2891.0	3039.0	92.67	0.790
Shannon	4.78	4.90	4.77	5.00	0.08	0.639
Simpson	0.04	0.04	0.04	0.03	0.00	0.532

^a,b^ Means within a row with different superscripts differ (*p* < 0.05). ^1^ CON, control group; FA, 12 mmol/L fumarate was added; NG, 99 μmol/L nitroglycerin was added; FN, fumarate and nitroglycerin were added in combination (FA + NG, 99 μmol/L + 12 mmol/L).

**Table 5 biology-12-01011-t005:** The relative abundances of bacteria at the phylum level (the average relative abundances of phylum > 1% in at least one group are presented).

Items	Groups ^1^				SEM	*p*-Value
	CON	FA	NG	FN		
Bacteroidetes	39.10 ^ab^	45.33 ^ab^	25.68 ^b^	55.39 ^a^	3.84	0.015
Firmicutes	36.18 ^b^	30.31 ^b^	56.25 ^a^	30.05 ^b^	3.73	0.009
Proteobacteria	16.51 ^a^	16.77 ^a^	7.69 ^b^	6.36 ^b^	1.63	0.004
Spirochaetae	2.20 ^ab^	2.42 ^a^	0.48 ^b^	2.65 ^a^	0.32	0.026
Actinobacteria	1.93 ^b^	1.00 ^b^	6.54 ^a^	1.89 ^b^	0.73	0.004
Candidate_division_SR1	0.90	0.82	0.49	1.17	0.11	0.158
Candidate_division_TM7	1.14 ^a^	0.62 ^b^	0.56 ^b^	0.49 ^b^	0.09	0.006
Cyanobacteria	0.82	1.25	1.50	1.20	0.19	0.702

^a,b^ Means within a row with different superscripts differ (*p* < 0.05). ^1^ CON, control group; FA, 12 mmol/L fumarate was added; NG, 99 μmol/L nitroglycerin was added; FN, fumarate and nitroglycerin were added in combination (FA + NG, 99 μmol/L + 12 mmol/L).

**Table 6 biology-12-01011-t006:** The relative abundances of bacteria at the genus level (the average relative abundances of genus > 2% in at least one group are presented).

Items		Groups ^1^				SEM	*p*-Value
		CON	FA	NG	FN		
Phylum	Genus						
Bacteroidetes	*Prevotella*	22.14	27.05	14.46	34.5	2.84	0.053
	*Rikenellaceae_RC9_gut_group*	6.56	8.36	5.62	7.37	0.46	0.187
	Unclassified *Prevotellaceae*	3.77	3.35	2.14	6.71	0.72	0.075
	Unclassified *BS11_gut_group*	2.31 ^a^	2.58 ^a^	0.78 ^b^	0.96 ^b^	0.26	0.002
	Unclassified *S24-7*	1.81	0.68	1.32	2.27	0.25	0.108
	Unclassified *RF16*	1.78	2.13	1.03	2.43	0.24	0.192
Firmicutes	Unclassified *Ruminococcaceae*	7.62 ^a^	6.00 ^ab^	3.92 ^b^	4.26 ^ab^	0.55	0.030
	Unclassified *Christensenellaceae*	4.09	2.76	3.13	2.63	0.31	0.370
	*Streptococcus* ^2^	3.40 ^b^	2.20 ^b^	26.94 ^a^	5.69 ^b^	3.40	<0.050
	*Butyrivibrio*	2.80	2.00	3.18	2.20	0.36	0.689
	*Succiniclasticum*	2.20	2.43	1.12	1.48	0.21	0.068
	*Ruminococcus*	2.13	1.30	2.38	1.30	0.19	0.063
	*Quinella*	1.63	2.15	1.12	1.53	0.18	0.238
Proteobacteria	*Succinivibrio*	8.33 ^a^	8.94 ^a^	4.53 ^ab^	2.31 ^b^	0.98	0.017
	Unclassified *Succinivibrionaceae*	3.93	4.32	1.71	2.2	0.42	0.064
	*Ruminobacter*	3.77 ^a^	2.32 ^ab^	1.08 ^b^	1.48 ^b^	0.35	0.006
Spirochaetae	*Treponema*	2.18 ^ab^	2.38 ^ab^	0.47 ^b^	2.64 ^a^	0.32	0.027
Actinobacteria	*Bifidobacterium*	1.05 ^b^	0.48 ^b^	5.66 ^a^	1.46 ^b^	0.67	0.024

^a,b^ Means within a row with different superscripts differ (*p* < 0.05). ^1^ CON, control group; FA, 12 mmol/L fumarate was added; NG, 99 μmol/L nitroglycerin was added; FN, fumarate and nitroglycerin were added in combination (FA + NG, 12 mmol/L + 99 μmol/L). ^2^ *p* value, 0.0498.

**Table 7 biology-12-01011-t007:** The relative abundances of archaea at the order level (the average relative abundances of orders > 1% in at least one group are presented).

Items	Groups ^1^				SEM	*p*-Value
	CON	FA	NG	FN		
Methanobacteriales	99.57 ^a^	96.59 ^b^	98.43 ^a^	90.06 ^c^	1.13	<0.001
Methanomassiliicoccales	0.43 ^c^	3.41 ^b^	1.57 ^c^	9.94 ^a^	1.13	<0.001

^a–c^ Means within a row with different superscripts differ (*p* < 0.05). ^1^ CON, control group; FA, 12 mmol/L fumarate was added; NG, 99 μmol/L nitroglycerin was added; FN, fumarate and nitroglycerin were added in combination (FA + NG, 12 mmol/L + 99 μmol/L).

**Table 8 biology-12-01011-t008:** The relative abundances of archaea at the species level (the average relative abundances of species > 1% in at least one group are presented).

Items	Groups ^1^				SEM	*p*-Value
	CON	FA	NG	FN		
*Methanobrevibacter gottschalkii clade*	74.93 ^a^	68.46 ^ab^	68.11 ^ab^	61.89 ^b^	2.17	0.002
*Methanobrevibacter boviskoreani clade*	6.35	1.20	0.84	0.91	1.68	0.055
*Methanobrevibacter ruminantium clade*	13.47	20.28	23.88	22.49	2.53	0.082
*Methanosphaera* sp. *ISO3-F5*	4.02	6.47	5.46	4.60	0.65	0.268
*Group12* sp. *ISO4-H5*	0.27 ^b^	1.57 ^b^	0.78 ^b^	6.45 ^a^	1.08	0.019
*Group9* sp. *ISO4-G1*	0.06 ^c^	1.05 ^b^	0.40 ^c^	2.30 ^a^	0.37	<0.001

^a–c^ Means within a row with different superscripts differ (*p* < 0.05). ^1^ CON, control group; FA, 12 mmol/L fumarate was added; NG, 99 μmol/L nitroglycerin was added; FN, fumarate and nitroglycerin were added in combination (FA + NG, 12 mmol/L + 99 μmol/L).

## Data Availability

The data presented in this study are available from the corresponding author on reasonable request.

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
