# Peer review of "Effects of Fumarate and Nitroglycerin on In Vitro Rumen Fermentation, Methane and Hydrogen Production, and on Microbiota"

_biology, 2023, doi:10.3390/biology12071011_

Round 1
Reviewer 1 Report
Some of my observation related to the study is given below:
1. The title of the article gives the impression that the study is an in vivo experiment but it is not so, therefore the title may be changed to “Effects of fumarate and nitroglycerin on in vitro rumen fermentation, methane and hydrogen production, and microbiota’
2. The authors have not stated anywhere in the article the basis for selecting the doses of 0 and 12 mmol/L for Fumerate and 0 and 99 µmol/L for Nitroglycerine. This is very important and the basis for selecting the dose must be stated.
3. The methodology for volatile fatty acids determination using GC needs to be rewritten. Chromatographic conditions, column details, etc. must be mentioned in the manuscript.
The whole of the ‘Results’ section needs to be re-written considerably. Only the main effect needs to be mentioned with reference to the Table or Figure’.
Comments to the Author
Line 31: change ‘with the occurrence of’ to ‘and resulted in’
Line 52: change ‘plants’ to ‘coarse feedstuffs’
Line 58: change to ‘Strategies like feed additives’
Line 61: Scientific name must be in italic ‘Asparagopsis taxiformis’
Line 62: change to ‘developed as’
Line 88 to 89: Please state the basis for selecting the doses of 0 and 12 mmol/L for Fumerate and 0 and 99 µmol/L for Nitroglycerine?
Line 99: Change ‘incubator’ to ‘incubated’
Line 104: Change to ‘0.015g MnCl2·4H2O’
Line 123-124: Please name the ‘deproteinization acidification solution’
Line 119-126: The methodology for volatile fatty acids determination using GC needs to be rewritten. Please state the chromatographic conditions, column details, etc. in the article.
Line 154: Rewrite as ‘operational taxonomic unit (OTU’
Line 171: The whole of the ‘Results’ section needs to be re-written considerably. Only the main effect needs to be mentioned with reference to the Table or Figure’.
Line 173: Change ‘FA by NG’ to ‘their combination’
Line 179: Why methane production was higher in FA than the CON group? It should be otherwise??
Line 191: Change to ‘The in vitro fermentation characteristics is presented in Table 2.’
Line 191-192: Check the statement ‘The pH was higher in FA than CON (P < 0.05). There was no difference between other groups in pH (P > 0.05).’ for correctness. Refer to Table 2.
Line 173-307: Please re-write the entire results with reference to control and treatments. Please state if it is ‘higher’ or ‘lower’. And there is no need to give ‘P value’ if the treatments are similar.
Line 216-220. Why the population of Archaea was higher in FA group than the Control? It should have been otherwise? Please explain your finding??
Line 239-253: Please re-write the entire results with reference to control and treatments. Please state if it is ‘higher’ or ‘lower’. And there is no need to give ‘P value’ if the treatments are similar.
Line 259-273: Please re-write the entire results with reference to control and treatments. Please state if it is ‘higher’ or ‘lower’. And there is no need to give ‘P value’ if the treatments are similar.
Line 281: Please check the statement ‘A total of 389,535 archaeal sequences were obtained after quality filtering.’ with Table 4
Line 344-345: Change ‘made’ to ‘altered’; rewrite as ‘Firmicutes and changes…….’
Line 313-377: Please discuss the points raised in Line no. 179, 216-220,
Line 31: change ‘with the occurrence of’ to ‘and resulted in’
Line 52: change ‘plants’ to ‘coarse feedstuffs’
Line 58: change to ‘Strategies like feed additives’
Line 61: Scientific name must be in italic ‘Asparagopsis taxiformis’
Line 62: change to ‘developed as’
Line 88 to 89: Please state the basis for selecting the doses of 0 and 12 mmol/L for Fumerate and 0 and 99 µmol/L for Nitroglycerine?
Line 99: Change ‘incubator’ to ‘incubated’
Line 104: Change to ‘0.015g MnCl2·4H2O’
Line 123-124: Please name the ‘deproteinization acidification solution’
Line 119-126: The methodology for volatile fatty acids determination using GC needs to be rewritten. Please state the chromatographic conditions, column details, etc. in the article.
Line 154: Rewrite as ‘operational taxonomic unit (OTU’
Line 171: The whole of the ‘Results’ section needs to be re-written considerably. Only the main effect needs to be mentioned with reference to the Table or Figure’.
Line 173: Change ‘FA by NG’ to ‘their combination’
Line 179: Why methane production was higher in FA than the CON group? It should be otherwise??
Line 191: Change to ‘The in vitro fermentation characteristics is presented in Table 2.’
Line 191-192: Check the statement ‘The pH was higher in FA than CON (P < 0.05). There was no difference between other groups in pH (P > 0.05).’ for correctness. Refer to Table 2.
Line 173-307: Please re-write the entire results with reference to control and treatments. Please state if it is ‘higher’ or ‘lower’. And there is no need to give ‘P value’ if the treatments are similar.
Line 216-220. Why the population of Archaea was higher in FA group than the Control? It should have been otherwise? Please explain your finding??
Line 239-253: Please re-write the entire results with reference to control and treatments. Please state if it is ‘higher’ or ‘lower’. And there is no need to give ‘P value’ if the treatments are similar.
Line 259-273: Please re-write the entire results with reference to control and treatments. Please state if it is ‘higher’ or ‘lower’. And there is no need to give ‘P value’ if the treatments are similar.
Line 281: Please check the statement ‘A total of 389,535 archaeal sequences were obtained after quality filtering.’ with Table 4
Line 344-345: Change ‘made’ to ‘altered’; rewrite as ‘Firmicutes and changes…….’
Line 313-377: Please discuss the points raised in Line no. 179, 216-220,
Author Response
Response to Reviewer 1 Comments
Dear reviewers:
We would like to thank you for the time and considerable effort given to the review of this manuscript. You will find that we have addressed all of the comments and made significant revisions to the manuscript “Effects of Fumarate and Nitroglycerin on in Vitro Rumen Fermentation, Methane and Hydrogen Production, and on Microbiota” (Manuscript ID: 2443381). We hope that you find the revised manuscript to be greatly improved and comments to be adequately addressed. Revised portion are marked up using the “highlighted” function in the manuscript.
Some of my observation related to the study is given below:
1.The title of the article gives the impression that the study is an in vivo experiment but it is not so, therefore the title may be changed to “Effects of fumarate and nitroglycerin on in vitro rumen fermentation, methane and hydrogen production, and microbiota”
Response: Thanks very much for this suggestion. Combining the comment of reviewer 5, the title has been revised to “Effects of Fumarate and Nitroglycerin on in Vitro Rumen Fermentation, Methane and Hydrogen Production, and on Microbiota” in line 2-3.
- The authors have not stated anywhere in the article the basis for selecting the doses of 0 and 12 mmol/L for Fumerate and 0 and 99 µmol/L for Nitroglycerine. This is very important and the basis for selecting the dose must be stated.
Response: Thanks very much for this comment. The basis has been given in the manuscript “The dosages of FN and NG used in this study were determined according to the previous studies [19, 30] and the results of the pre-experiments.” in line 96-98.
- The methodology for volatile fatty acids determination using GC needs to be rewritten. Chromatographic conditions, column details, etc. must be mentioned in the manuscript.
The whole of the ‘Results’ section needs to be re-written considerably. Only the main effect needs to be mentioned with reference to the Table or Figure’.
Response: Thanks very much for this suggestion.
1) The methodology for volatile fatty acids determination has been revised according to the comment in line 132-138.
2) The results have been rewritten according to the comment in line 187-191, 197-207, 214-216, 239-244, and 255-260.
Comments to the Author
Line 31: change ‘with the occurrence of’ to ‘and resulted in’
Response: Thanks very much for this suggestion. It has been revised in line 32.
Line 52: change ‘plants’ to ‘coarse feedstuffs’
Response: Thanks very much for this suggestion. It has been revised in line 53.
Line 58: change to ‘Strategies like feed additives’
Response: Thanks very much for this suggestion. It has been revised in line 59.
Line 61: Scientific name must be in italic ‘Asparagopsis taxiformis’
Response: Thanks very much for this suggestion. It has been revised in line 63.
Line 62: change to ‘developed as’
Response: Thanks very much for this suggestion. It has been revised in line 63.
Line 88 to 89: Please state the basis for selecting the doses of 0 and 12 mmol/L for Fumerate and 0 and 99 µmol/L for Nitroglycerine ?
Response: Thanks very much for this comment. The basis has been given in the manuscript “The dosages of FN and NG used in this study were determined according to the previous studies [19, 30] and the results of the pre-experiments.” in line 96-98.
Line 99: Change ‘incubator’ to ‘incubated’
Response: Thanks very much for this suggestion. It has been revised in line 107.
Line 104: Change to ‘0.015g MnCl2·4H2O’
Response: Thanks very much for this suggestion. It has been revised in line 111.
Line 123-124: Please name the ‘deproteinization acidification solution’
Response: Thanks very much for this suggestion. More information has been given for the solution “Each 1.0 mL sample was mixed with 0.2 mL deproteinization–acidification solution [metaphosphoric acid (25% w/v) and crotonic acid (0.65% w/v)] before undergoing analy-sis via gas chromatography (Agilent 7890B instrument, Agilent, California, USA).” in line 132-134.
Line 119-126: The methodology for volatile fatty acids determination using GC needs to be rewritten. Please state the chromatographic conditions, column details, etc. in the article.
Response: Thanks very much for this comment. The methodology for volatile fatty acids determination has been revised according to the comment in line 132-138.
Line 154: Rewrite as ‘operational taxonomic unit (OUT)’
Response: Thanks very much for this comment. It has been revised to operational taxonomic unit (OTU) in line 168.
Line 171: The whole of the ‘Results’ section needs to be re-written considerably. Only the main effect needs to be mentioned with reference to the Table or Figure’.
Response: Thanks very much for this comment. The results have been rewritten according to the comment in line 187-191, 197-207, 214-216, 239-244, and 255-260.
Line 173: Change ‘FA by NG’ to ‘their combination’
Response: Thanks very much for this suggestion. After careful consideration , we replaced “FA by NG” with “FN”, and have rewritten the results in line 187-191.
Line 179: Why methane production was higher in FA than the CON group? It should be otherwise??
Line 216-220. Why the population of Archaea was higher in FA group than the Control? It should have been otherwise? Please explain your finding??
Line 313-377: Please discuss the points raised in Line no. 179, 216-220
Response: Thanks very much for this comment. More discussion has been given. “Fumarate increased the abundance of archaea, methane production and acetate with the absence of nitroglycerin. Fumarate was expected to consume H2 and reduce methane production, the increase of methane production was not expected. Fumarate can be metabolized into acetate via malate–pyruvate pathway in the rumen [19]. In this process, there is net [H] produced (C4H4O4 + 2H2O→C2H4O2 + 2CO2 + 4H), which could account for the increase in the abundance of archaea and methane production. The increased concentration of acetate supported this speculation. Moreover, Gibbs free energy calculation shows that the production of acetate from fumarate under rumen conditions is thermodynamically feasible even at very low fumarate concentration [35]. Fumarate was metabolized into acetate instead of propionate, which could have occurred as the microbial populations that reduce fumarate to succinate/propionate had been yet completely established.” in line 328-339.
Line 191: Change to ‘The in vitro fermentation characteristics is presented in Table 2.’
Response: Thanks very much for this suggestion. We have replaced “The fermentation characteristics was shown in Table 2.” with “The in vitro fermentation characteristics are presented in Table 2.” in line 197.
Line 191-192: Check the statement ‘The pH was higher in FA than CON (P < 0.05). There was no difference between other groups in pH (P > 0.05).’ for correctness. Refer to Table 2.
Response: Thanks very much for this comment. It was our mistake. It has been revised to “There were no significant differences in pH, butyrate and lactate among the 4 groups (P > 0.05).” in line 206.
Line 173-307: Please re-write the entire results with reference to control and treatments. Please state if it is ‘higher’ or ‘lower’. And there is no need to give ‘P value’ if the treatments are similar.
Response: Thanks very much for this comment. According to the comments, the “Results” was rewritten in line 187-191, 197-207, 214-216, 239-244, and 255-260.
Line 239-253: Please re-write the entire results with reference to control and treatments. Please state if it is ‘higher’ or ‘lower’. And there is no need to give ‘P value’ if the treatments are similar.
Response: Thanks very much for this comment. According to the comments, the “Results” was rewritten in line 187-191, 197-207, 214-216, 239-244, and 255-260.
Line 259-273: Please re-write the entire results with reference to control and treatments. Please state if it is ‘higher’ or ‘lower’. And there is no need to give ‘P value’ if the treatments are similar.
Response: Thanks very much for this comment. According to the comments, the “Results” was rewritten in line 187-191, 197-207, 214-216, 239-244, and 255-260.
Line 281: Please check the statement ‘A total of 389,535 archaeal sequences were obtained after quality filtering.’ with Table 4
Response: Thanks very much for this comment. It was our mistake. It has been revised to “568, 155 archaeal sequences” in line 268.
Line 344-345: Change ‘made’ to ‘altered’; rewrite as ‘Firmicutes and changes…….’
Response: Thanks very much for the comments. It has been revised according to the comments in line 341-342.

Reviewer 2 Report
This manuscript presents results from an in vitro study assessing the methane-reducing potential of fumarate, nitroglycerin, or their combination during incubation of rumen microbes. The manuscript is generally well written. The topic is appropriate for the journal. The authors provide a good introduction describing the background and rationale for their study. their experimental objectives are clearly stated. Their materials and methods are described concisely and clearly and are appropriate to achieve stated objectives. The results are presented and discussed clearly and objectively. conclusions reached are supported by evidence. I list below only 1 minor comment:
line 61; italicize latinized genus and species names.
Author Response
Response to Reviewer 2 Comments
Dear reviewers:
We would like to thank you for the time and considerable effort given to the review of this manuscript. You will find that we have addressed all of the comments and made significant revisions to the manuscript “Effects of fumarate and nitroglycerin on in vitro rumen fermentation, methane and hydrogen production, and on microbiota” (Manuscript ID: 2443381). We hope that you find the revised manuscript to be greatly improved and comments to be adequately addressed. Revised portion are marked up using the “highlighted” function in the manuscript.
This manuscript presents results from an in vitro study assessing the methane-reducing potential of fumarate, nitroglycerin, or their combination during incubation of rumen microbes. The manuscript is generally well written. The topic is appropriate for the journal. The authors provide a good introduction describing the background and rationale for their study. their experimental objectives are clearly stated. Their materials and methods are described concisely and clearly and are appropriate to achieve stated objectives. The results are presented and discussed clearly and objectively. conclusions reached are supported by evidence. I list below only 1 minor comment:
line 61; italicize latinized genus and species names.
Response: Thank you very much for this suggestion. It has been revised “Asparagopsis taxiformis” in line 63.

Reviewer 3 Report
General comments
-In L18 “nitroglycerin inhibits the activities of methanogens” but Archaea and methanogens did not affect by NG supplementation when compared with control (Table 3,7,8).
Discussion: For most of the discussion, you wrote fumarate. Please rewrite and discuss more about nitroglycerin in rumen fermentation and methanogenesis.
Specific comments
L26: Changed to “….of fumarate and nitroglycerin on the rumen fermentation, methane, and hydrogen production, and microbiota.”
L38: Changed to “Unclassified Ruminococcaceae…...” and in Table 6.
L88: 1) Why you use 2x2 factorial design. Normally, your experimental design must use CRD for 4 treatment 2) Changed to “Control (CON), fumarate at 12 mmol/L (FA), nitroglycerin at 12 mmol/L (NG), and fumarate at 12 mmol/L (FA) plus nitroglycerin at 12 mmol/L (FN).”
L95: When you sampling rumen fluid after you raised sheep, such as “The animals were fed the diets for a period of 14 days. On day 15, 500 mL of rumen fluid was sampled…”
L173: Changed to “…, and FN in the total gas…..”
Table 1: When you added NG and FN, they could produce methane at 0 mL (which, in my opinion, is impossible). Normally, microbes produce methane in the rumen fermentation. But in Tables 3, 7 and 8, the populations of archaea and methanogens were shown. So, methane production and archaea and methanogen populations have no relationship. Please check the data.
Must be improved
Author Response
Response to Reviewer 3 Comments
Dear reviewers:
We would like to thank you for the time and considerable effort given to the review of this manuscript. You will find that we have addressed all of the comments and made significant revisions to the manuscript “Effects of fumarate and nitroglycerin on in vitro rumen fermentation, methane and hydrogen production, and on microbiota” (Manuscript ID: 2443381). We hope that you find the revised manuscript to be greatly improved and comments to be adequately addressed. Revised portion are marked up using the “highlighted” function in the manuscript.
General comments
-In L18 “nitroglycerin inhibits the activities of methanogens” but Archaea and methanogens did not affect by NG supplementation when compared with control (Table 3,7,8).
Response: Thanks for the comments. A discussion has been given. “The current experiment also observed that nitroglycerin inhibited methane production but did not affect the abundance of archaea compared with the control. This result is consistent with that of a previous study [30]. In another study, an opposite result was observed, whereby the abundance of archaea declined when the methane production was depressed by nitroglycerin [17]. This study was consistent with the research on the other methanogenic inhibitors [31, 32]. However, the mechanisms of the different results for archaeal abundance in different studies are unclear and further work is needed to elucidate this point.” in line 309-316.
Discussion: For most of the discussion, you wrote fumarate. Please rewrite and discuss more about nitroglycerin in rumen fermentation and methanogenesis.
Response: Thanks very much for this suggestion. We have rewritten and discussed more about nitroglycerin in rumen fermentation and methanogenesis. “Nitroglycerin was demonstrated to be effective at reducing rumen methane production in several in vitro and in vivo studies [17, 18, 30]. It was able to completely inhibit methane production and caused an accumulation of hydrogen in in vitro rumen fermentation [17, 30]. Moreover, the final metabolites of nitroglycerin were propionate and ammonia in the rumen, which have no negative effect on rumen fermentation. Therefore, nitroglycerin was used to successfully establish a model of methane depression and hydrogen accumulation in this study. Nitroglycerin caused about 4.8% hydrogen of accumulation (% total gas production). The current experiment also observed that nitroglycerin inhibited methane production but did not affect the abundance of archaea compared with the control. This result is consistent with that of a previous study [30]. In another study, an opposite result was observed, whereby the abundance of archaea declined when the methane production was depressed by nitroglycerin [17]. This study was consistent with the research on the other methanogenic inhibitors [31, 32]. However, the mechanisms of the different results for archaeal abundance in different studies are unclear and further work is needed to elucidate this point.” in line 302-316.
Specific comments
L26: Changed to “…of fumarate and nitroglycerin on the rumen fermentation, methane, and hydrogen production, and microbiota.”
Response: Thanks very much for this suggestion. It has been revised to “This study aimed to investigate the effects of fumarate and nitroglycerin on the rumen fermentation, methane, and hydrogen production, and microbiota.” in line 27-28.
L38: Changed to “Unclassified Ruminococcaceae…...” and in Table 6.
Response: Thanks very much for this suggestion. It has been revised to “Unclassified Ruminococcaceae” in L39 and in Table 6.
L88: 1) Why you use 2x2 factorial design. Normally, your experimental design must use CRD for 4 treatment 2) Changed to “Control (CON), fumarate at 12 mmol/L (FA), nitroglycerin at 12 mmol/L (NG), and fumarate at 12 mmol/L (FA) plus nitroglycerin at 12 mmol/L (FN).”
Response: Thanks very much for the comments. We agree with you. It has been revised to “Rumen in vitro fermentation was carried out with a completely random design for 4 treatments: control (CON), fumarate at 12 mmol/L (FA), nitroglycerin at 99 μmol/L (NG), and fumarate at 12 mmol/L plus nitroglycerin at 99 μmol/L (FN).” in line 94-96.
L95: When you sampling rumen fluid after you raised sheep, such as “The animals were fed the diets for a period of 14 days. On day 15, 500 mL of rumen fluid was sampled…”
Response: Thanks very much for the comments. It has been revised. “Three rumen-fistulated Chinese Hu sheep were fed on a maintenance diet for a period of 30 days. On day 31, 500 mL of rumen fluid was collected 2 h before the sheep were fed.” in line 81-82.
L173: Changed to “…, and FN in the total gas…..”
Response: Thanks very much for this suggestion. Taking the comments of other reviewers into consideration, the results section has been rewritten.
Table 1: When you added NG and FN, they could produce methane at 0 mL (which, in my opinion, is impossible). Normally, microbes produce methane in the rumen fermentation. But in Tables 3, 7 and 8, the populations of archaea and methanogens were shown. So, methane production and archaea and methanogen populations have no relationship. Please check the data.
Response: Thanks very much for the comment. Under the experimental condition of this study, methane was not detected by addtion of NG.Three independent incubation conducted at different time showed the same results. Similar results were also observed in a previous study by adding NG [30]. A discussion has been given. “The current experiment also observed that nitroglycerin inhibited methane production but did not affect the abundance of archaea compared with the control. This result is consistent with that of a previous study [30]. In another study, an opposite result was ob-served, whereby the abundance of archaea declined when the methane production was depressed by nitroglycerin [17], which was consistent with the researches on the other methanogenic inhibitors [31, 32]. However, the mechanisms of the different results for archaeal abundance in different studies are unclear and further work is needed to elucidate this point.” in line 309-316.

Reviewer 4 Report
|
Lines |
Comment |
|
12-13 |
Rewrite so that it is one thought so that it reads easier. |
|
50 |
Change “the” to “a” remove “feed” at end of sentence. |
|
53 |
Change “was” to “is” |
|
55 |
Change “would” to “could” |
|
56 |
Change to “fermentation, therefore” |
|
67 |
Remove “Therefore” and start the sentence with Fumarate. |
|
68 |
Remove the sentence starting with “Nitroglycerin…..in this study.” This is part of your objective not the introduction. |
|
69 |
Put the sentence starting with “Its metabolic……rumen” to the end of the paragraph. |
|
71 |
Change “it” to “which” |
|
76 |
Change “would” to “could” |
|
88-90 |
This is confusing the way it is written. It will read easier if you say control (CON), fumarate 12 mmol/L (FA), nitroglycerin 99 µmol/L (NG) and nitroglyrein99 µmol/L plus fumarate 12 mmol/L (FN). |
|
105 |
Change “to” to “as” |
|
123 |
Please state what the deproteinization acid was and what the concentration of acid was. |
|
127 |
Do not start a sentence with an abbreviation. Change MCP to what it stands for. |
|
In the statistical analysis |
What did you use to separate the means, pdiff, tukeys, etc. Also did you try transforming your non-normal variables and then run Mixed on them. |
|
173-174 |
Remove the sentence because it makes it seem like there are effects of all treatments on gas production. Start with “The total gas…” in line 174, this is similar to how you started your other results sections. |
|
228 |
State that this is for the bacteria only since there were no differences for Archaea |
|
253 |
Remove “there was” to “showed” |
|
315 |
Change “It would” to “This can” |
|
376 |
Change “It” to “This” |
Minor changes needed
Author Response
Response to Reviewer 4 Comments
Dear reviewers:
We would like to thank you for the time and considerable effort given to the review of this manuscript. You will find that we have addressed all of the comments and made significant revisions to the manuscript “Effects of fumarate and nitroglycerin on in vitro rumen fermentation, methane and hydrogen production, and on microbiota” (Manuscript ID: 2443381). We hope that you find the revised manuscript to be greatly improved and comments to be adequately addressed. Revised portion are marked up using the “highlighted” function in the manuscript.
12-13 Rewrite so that it is one thought so that it reads easier.
Response: Thanks very much for this comment. It has been rewritten “An important strategy to mitigate global warming is that reducing methane produced by ruminants. However, the inhibition of rumen methanogenesis generally results in hydrogen accumulation, which would affect the normal fermentation in the rumen.” in line 13-15.
50 Change “the” to “a” remove “feed” at end of sentence.
Response: Thanks very much for this comment. It has been revised in line 51.
53 Change “was” to “is”
Response: Thanks very much for this comment. It has been revised in line 54.
55 Change “would” to “could”
Response: Thanks very much for this comment. It has been revised in line 57.
56 Change to “fermentation, therefore”
Response: Thanks very much for this comment. It has been revised in line 57.
67 Remove “Therefore” and start the sentence with Fumarate.
Response: Thanks very much for this comment. It has been revised in line 69.
68 Remove the sentence starting with “Nitroglycerin…..in this study.” This is part of your objective not the introduction.
Response: Thanks very much for this comment. “Nitroglycerin was used to inhibit the activities of methanogens in this study.” has been removed.
69 Put the sentence starting with “Its metabolic……rumen” to the end of the paragraph.
Response: Thanks very much for this comment. It has been revised in line 73.
71 Change “it” to “which”
Response: Thanks very much for this comment. It has been revised in line 71.
76 Change “would” to “could”
Response: Thanks very much for this comment. It has been revised in line 77.
88-90 This is confusing the way it is written. It will read easier if you say control (CON), fumarate 12 mmol/L (FA), nitroglycerin 99 µmol/L (NG) and nitroglyrein 99 µmol/L plus fumarate 12 mmol/L (FN).
Response: Thanks very much for this comment. It has been revised in line 94-96.
105 Change “to” to “as”
Response: Thanks very much for this comment. It has been revised in line 113.
123 Please state what the deproteinization acid was and what the concentration of acid was.
Response: Thanks very much for this comment. It has been revised “Each 1.0 mL sample was mixed with 0.2 mL deproteinization–acidification solution [metaphosphoric acid (25% w/v) and crotonic acid (0.65% w/v)] before undergoing analy-sis via gas chromatography (Agilent 7890B instrument, Agilent, California, USA).” in line 132-134.
127 Do not start a sentence with an abbreviation. Change MCP to what it stands for.
Response: Thanks very much for this comment. It has been revised to “Microbial crude protein” in line 140.
In the statistical analysis
What did you use to separate the means, pdiff, tukeys, etc. Also did you try transforming your non-normal variables and then run Mixed on them.
Response: Thanks very much for the comments.
1)Tukey test was used to separate the means, it has been given in line 183-184.
2)It is a very good suggestion. But we did not transform non-normal variables and then run Mixed on them in this study. We will try in the following study.
173-174 Remove the sentence because it makes it seem like there are effects of all treatments on gas production. Start with “The total gas…” in line 174, this is similar to how you started your other results sections.
Response: Thanks very much for the comments. It has been revised to start with “The total gas”. And
the whole of “Results” has been rewritten in line 187-191, 197-207, 214-216, 239-244, and 255-260.
228 State that this is for the bacteria only since there were no differences for Archaea
Response: Thanks very much for the comments. It has been revised.
1) “There was a clear separation of clusters on the 3D-PCoA plot of bacterial populations among the 4 groups (Anosim, R=0.747, P=0.001, Figure 1 A).” in line 226;
2)“There was a clear separation of clusters on the 3D-PCoA plot of archaeal populations among the 4 groups (Anosim, R=0.410, P=0.005, Figure 1 B).” in line 271.
253 Remove “there was” to “showed”
Response: Thanks very much for this suggestion. it has been revised in line 244.
315 Change “It would” to “This can”
Response: Thanks very much for this suggestion. it has been revised in line 299.
376 Change “It” to “This”
Response: Thanks very much for this suggestion. it has been revised in line 372.

Reviewer 5 Report
In this manuscript, the authors describe the effects of fumarate on rumen fermentation, methane and hydrogen production, and microbiota when the methanogenesis was inhibited by nitroglycerin. The research is well-designed, and the results are presented satisfactorily. However, minor revisions are required in the manuscript. Detailed comments are given on the text.

Author Response
Response to Reviewer 5 Comments
Dear reviewers:
We would like to thank you for the time and considerable effort given to the review of this manuscript. You will find that we have addressed all of the comments and made significant revisions to the manuscript “Effects of fumarate and nitroglycerin on in vitro rumen fermentation, methane and hydrogen production, and on microbiota” (Manuscript ID: 2443381). We hope that you find the revised manuscript to be greatly improved and comments to be adequately addressed. Revised portion are marked up using the “highlighted” function in the manuscript.
In this manuscript, the authors describe the effects of fumarate on rumen fermentation, methane and hydrogen production, and microbiota when the methanogenesis was inhibited by nitroglycerin. The research is well-designed, and the results are presented satisfactorily. However, minor revisions are required in the manuscript. Detailed comments are given on the text.
Line 2 on microbiota?
Response: Thanks very much for this suggestion. it has been revised in line 2-3.
Avoid the use of personal pronouns like "we" and "our" when possible. This is a scientific article, not a novel or story! in line 73.
Response: Thanks very much for this suggestion. it has been revised throughout the manuscript in line 74.
Line which additives please be clear in line 83.
Response: Thanks very much for this suggestion. The information has been given “(Vitamin and mineral mix contained the following ingredients per kilogram of diet: vita-min A, 22.5 KIU/kg; vitamin D3, 5.0 KIU/kg; vitamin E, 37.5 IU/kg; vitamin K3, 5.0 mg/kg; Mn, 63.5 mg/kg; Zn, 111.9 mg/kg; Cu, 25.6 mg/kg; and Fe, 159.3 mg/kg.)” in line 86-88.
was the diet offered as a Total mixed ration?
Response: The information has been given “Sheep were fed a total mixed ration twice daily (08:00 and 17:00) and had free access to fresh water.” in line 90-92.
Is there any information available from your potential experience about what kind of side effects these levels you have used may have on the animal?
Response: Thanks very much for this comment. In our previous study, 100 mg/head/day nitroglycerin was given to Hu sheep, and got about 20% reduction of methane emission. The growth performance of sheep was not affected at this dosage of nitroglycerin. Fumarate has been widely studied on rumen methane mitigation by in vivo and in vitro studies, it was a relatively safe chemical. An in vitro trial is only to investigate the potential feasibility of this methane mitigation approach. Further animal studies must be carried out to detemine the dosage and side effects on animals.
There is an environment in which we do not know what it is like to carry out the positive results of this additive information without testing it in vivo. As a matter of fact, I believe that the development of modelling that can predict possible consequences on living material can provide insights for a better future in this field.
Response: Thanks very much for this comment. We agree with you. We learn a lot from these comments, it will help us in the futher study.
Informed Consent Statement:
Response: We have added relevant content “Informed Consent Statement: Not applicable.” in line 398.

Round 2
Reviewer 3 Report
L94: Changed to " completely randomized design (CRD)
L94: Changed to " completely randomized design (CRD)